# Canine distemper in Nepal's Annapurna Conservation Area – Implications of dog husbandry and human behaviour for wildlife disease

Debby Ng[1]*, Scott Carver[1], Mukhiya Gotame[2], Dibesh Karmasharya[3], Dikpal Karmacharya[4], Saman Man Pradhan[3], Ajay Narsingh Rana[2], Christopher N. Johnson[1]

**1** School of Natural Sciences, University of Tasmania, Hobart, Australia, **2** Himalayan Mutt Project, Kathmandu, Nepal, **3** Center for Molecular Dynamics, Kathmandu, Nepal, **4** Third Pole Conservancy, Kathmandu, Nepal

\* debby.ng@utas.edu.au

**Data Availability Statement:** All relevant data are within the manuscript and its Supporting Information files.

## Abstract

Dogs are often commensal with human settlements. In areas where settlements are adjacent to wildlife habitat, the management of dogs can affect risk of spillover of disease to wildlife. We assess dog husbandry practices, and measure the prevalence of Canine Distemper Virus (CDV) in dogs, in 10 villages in Nepal's Annapurna Conservation Area (ACA), an important region for Himalayan wildlife. A high proportion (58%) of owned dogs were allowed by their owners to roam freely, and many village dogs originated from urban areas outside the region. CDV antibodies, indicating past exposure, were detected in 70% of dogs, and 13% were positive for P-gene, suggesting current circulation of CDV. This is the first detection of canine distemper virus in a National Park in Nepal Himalaya. Dogs were generally in good condition, and none exhibited clinical signs of CDV infection, which suggests that infections were asymptomatic. CDV exposure varied with village location and age of dogs, but this variation was minor, consistent with high rates of movement of dogs across the region maintaining high seroprevalence. Residents reported the occurrence of several species of wild carnivores in or close to villages. These results suggest a high potential for transmission of CDV from village dogs to wild carnivores in ACA. We suggest that control of dog immigration, along with vaccination and neutering of dogs could mitigate the risk of CDV spillover into wild carnivore populations.

## Introduction

Domestic dogs *Canis lupus familiaris* (hereafter "dogs") have followed the human journey [1] to become the world's most abundant and widely distributed carnivore [2–4]. Dog husbandry refers to the selection and management of dogs (through roaming restrictions, feed type and frequency, veterinary care, reproductive management, and so on) to fulfil their assigned

**Funding:** The field work component of this study was funded by an Early Career Grant (#WW-191ER-17) from the National Geographic Society that was awarded to DN. The funders had no role in study design, data collection and analysis, decision to publish, or preparation of the manuscript. URL: https://www.nationalgeographic.org/find-explorers/debby-ng

**Competing interests:** The authors have declared that no competing interests exist.

function (i.e. pet, herding, guarding). Husbandry shapes the health and fitness of dogs, and affects their behaviour and physiology [5]. Through the process of deciding which dogs receive care, which get to reproduce, or even which live or die, humans shape the demography of dog populations [6]. Human behaviour also shapes interactions between dogs and wildlife [7]. Combined, dog husbandry and human behaviour shape the way dogs respond to pathogens as host or reservoir [8]. Studies have investigated the diseases shared between dogs and wildlife [9–11], but there have been few attempts to understand the importance of humans in facilitating the role of dogs as agents of pathogen transmission to wildlife [7,11,12].

Canine distemper virus (CDV) is a globally-distributed infectious disease that was once thought to affect dogs only. It has now been detected in over 200 species of terrestrial vertebrates [13,14], and remains an emerging disease because high mutability and subsequent host-switching enables the virus to emerge in previously unknown host species [15–17]. Although CDV can persist in wild carnivore populations other than dogs, dogs are considered its primary reservoir [18,19].

CDV has been associated with sudden and high mortalities that have decimated wild populations of threatened black-footed ferrets [20], Serengeti lions [21], and Ethiopian wolves [22]. No treatments exist for CDV, and vaccination of dogs remains the best prevention against infection in both dogs and wildlife [23]. Vaccinations have been demonstrated to be safe and effective for some species of captive wild carnivores in clinical trials [24–27], but in species native to the Himalaya such as snow leopards *Panthera uncia* and red pandas *Ailurus fulgens*, vaccines have been unreliable [28], or remain in trial [29]. Uncertainty surrounding the safety of CDV vaccines in wildlife hinders its use in conservation efforts, particularly in threatened or endangered species for which clinical trials are difficult. These features of the epidemiology of CDV could have far-reaching and devastating consequences for wild carnivores [10,19,30].

Annapurna Conservation Area (ACA), is Nepal's oldest, largest (7,629 km$^2$) and most visited conservation area, attracting 60 per cent of adventure tourists who travel to Nepal. It is home to 105,424 people who have built an economy on subsistence farming, pastoralism, and increasingly, tourism. Over 100 species of mammals have been recorded in ACA including species susceptible to CDV such as yellow-throated marten *Martes flavigula*, red fox *Vulpes vulpes*, golden jackal *Canis aureus* [31], Pallas' cat *Otoc olobus manul*, Himalayan wolf *Canis lupus filchneri*, common leopard *Panthera pardus*, and snow leopard *Panthera uncia* [32]. Communities in this region have traditionally employed dogs for guarding and herding. Despite this long-held association, studies of dogs in the Himalaya have been limited [33–35], and none have investigated the culture of dog husbandry among Himalayan communities, let alone the implications of this for the health of dogs. The recent development of highways that connect urban centres like Kathmandu with Himalayan villages has dramatically reduced time needed for residents the travel between villages and urban centres [36]. Prior to 2007, access to the study area was limited to foot access only. Mountain flights were available but used almost exclusively by tourists as high costs were prohibitive for residents. Today, several large off-road vehicles arrive at the study area daily, ferrying people, their goods and their animals. Despite these developments, veterinary services remain absent and have resulted in a large majority of dogs remaining unvaccinated. The confluence of these actors and processes make it an opportune time to investigate the interactions of people, dogs, wildlife and disease in ACA.

We describe the culture of dog husbandry in ACA and gather data on dog demography using a household questionnaire survey. Because host abundance is an important characteristic in disease maintenance [37–39], we focussed on the relationships between household variables and husbandry practices and the number of dogs in each household. We also determined the seroprevalence of CDV among dogs in the ACA and studied the variables (S1 Table) that

might be useful in predicting CDV exposure in individual dogs. Assessing the prevalence of CDV among dogs in ACA will be essential for assessing if there is a risk of disease transmission from dogs to sympatric wildlife. To gather insight on dog-wildlife interactions, our questionnaire included reports of observations of potentially susceptible wildlife species near villages and instances of dog-wildlife-livestock conflict.

## Materials and methods

### Study area and sampling design

This study was conducted in ten villages located in Nepal's Annapurna Conservation Area (ACA) (28˚40'13.3"N 83˚58'17.2"E to 28˚34'22.3"N 84˚11'24.3"E). The following villages were visited over four weeks in May 2018: Khangsar (4,200 m above sea level), Tanki Manang (3,683 m a.s.l.), Manang (3,558 m a.s.l.), Bhraka (3,542 m a.s.l.), Munji (3,454 m a.s.l.), Ngarwal (3,700 m a.s.l.), Humdre (3,415 m a.s.l.), Ghyaru (3,720 m a.s.l.), Pisang (3,200 m a.s.l.), and Bhratang (2,902 m a.s.l.). The study area was approximately 570 km$^2$ and comprised 10 villages with a human population of approximately 1,900 (34% of the total population of the ACA), with an average of 2.7 people per household [40].

We approached every dog owning household (DOHH) across the 10 villages. Permission to participate in the study was sought from the dog's owner. DOHH were identified through information supplied by the mayor of each village. Only dogs with owners who agreed to respond to the questionnaire were included in the serological analyses. Dogs were considered "owned" if: the dog was fed daily by the same individual (or persons strongly associated with that individual, i.e. a family), and if the same individual provided for its basic healthcare (e.g. tending to wounds, fending off aggressive dogs, provided the dog a safe space to feed and rest, etc.). One dog owner in Manang refused to respond to the questionnaire and was thus excluded from the study. Two dogs from separate owners, also in Manang, were excluded from the study as both dogs were old and did not respond well to attempts to collect blood. To avoid further distress to the dog owners and to ensure welfare of the dogs, attempts at blood collection were halted after a few tries. The owners of both dogs responded to the questionnaire but these responses were excluded from analyses due of an absence of serological data. A total of 76 households were interviewed, representing 12.8% of the total number of households in the study area and 96.2% of all DOHH. Responses from 71 households were subsequently included in the analyses. The village dog population is well-represented by these 71 dogs as owned dogs comprised the majority of dogs across the study site (only 15 dogs did not fit our criteria of having owners).

### Questionnaire

A questionnaire was developed to obtain detailed information on the demography of residents and their attitudes towards and relationships to domestic dogs (S1 Appendix). The questionnaire was modelled on guidelines from the World Health Organisation [41] and similar published studies [42]. The questionnaire sought information about the household, and the individual dog/s associated with each household. The main data fields at the household level were: number of people; number of children; number of dogs; frequency and type of food fed to dogs; and type and quantity of livestock owned (which was later used as a proxy for household income). Participants were asked to report the births and mortality of dogs in the household in the past 12 months. Data fields relating to individual dogs were: dog function; breed; roaming behaviour; sterilisation status; vaccination history; age; sex; and source/origin. As a measure of care afforded to dogs, owners were asked if their dog had visited a veterinarian or been dewormed within the past 12 months.

Interviewees were asked to report livestock that had been killed by wildlife in the past 12 months, and to identify, by pointing to photographs (Fig in S1 Appendix) supplied with the questionnaire, native carnivores (yellow-throated marten *Martes flavigula*, snow leopard *Panthera uncia*, golden jackal *Canis aureus*, Himalayan wolf *Canis lupus filchneri*, red fox *Vulpes vulpes*) they had observed in proximity to their property and/or dogs. A random number generator was used to vary the order in which the options were presented, and thus to remove effects of choice order on aggregated responses. A random number generator was also used to pair the questionnaire with the household.

Before administrating the questionnaire, it was pre-tested with members of the Manang community that resided outside the study area. The purposes of the pre-test were to (i) develop consistency in the duration required to administer each questionnaire, and (ii) test the efficacy of the questionnaire, which was designed in English and translated into Nepali. The latter was achieved by back-translating questions and responses to determine if desired questions were being received by interviewees, and if responses were accurately translated. All interviews were carried out in Nepali by the same interviewer.

A Garmin ® Edge 705 was used to record the latitude and longitude for each household. Based on this location, altitude was extracted from a 90M digital elevation model of Nepal, sourced from the Humanitarian Data Exchange (https://data.humdata.org/dataset/nepal-digital-model-elevation-dem).

## Determining CDV seroprevalence and body condition

After completion of the questionnaire, dog owners were asked for permission to have blood samples collected from their dogs. The serosurvey included only dogs that had no known vaccination against CDV, as attested by their owners. After blood collection, dogs were marked with a non-toxic red crayon on the forehead to ensure that they were not sampled twice. Dogs of all ages and body conditions were sampled. Where there were litters of pups below 12 weeks, only one individual was sampled as exposure to CDV is likely identical for litter-mates [43].

Upon receiving consent from the dog's owner, our veterinarian collected 2-3mL of whole blood from the cephalic vein with assistance of our veterinary technician and preserved the sample with EDTA. All tubes were refrigerated at 5˚C in the field before being delivered to the National Zoonoses and Food Hygiene Research Centre (NZFHRC) in Kathmamdu, where the blood samples were frozen after being portioned for ELISA assay and PCR analysis. The latter was stored in a Remel$^{TM}$ MicroTest$^{TM}$ M6$^{TM}$ Viral Transport Medium (VTM) before being delivered to the Centre for Molecular Dynamics (CMDN) in Kathmandu, Nepal. Of the blood collected from 71 dogs, 68 (95.8%) had sufficient sera for ELISA assay. Of the ELISA-positive samples, 48 (67.6%) had sufficient blood for for P-gene detection with PCR analysis.

Descriptions of clinical skin and/or body condition, and a Body Condition Score (BCS) were collected for each dog (S2 Appendix). Skin condition and BCS were judged by the same veterinarian for all dogs to eliminate observer bias. The presence of a clinical skin condition (e.g. squamous cell carcinoma, mange, etc.) was classified into one of two categories: <20%, or >20% coverage of the body. Body condition notes described physical abnormalities that included evidence of existing or past injury (e.g. open wounds, limp), and/or underlying disease (e.g. cysts, penile prolapse). The BCS is a semiquantitative, standard observational rating method for assessing body fat and muscle mass [44,45]. A 9-point BCS scale was used in which the midrange represents optimal body condition, lower values represent lean to emaciated conditions, and higher values indicate excessive body fat. All dogs involved in the survey were photographically identified.

Seropositivity to CDV was determined by an ELISA assay (Demeditec CDV (Canine Distemper Virus) IgG ELISA DE2478 Demeditec Diagnostics GmbH, Kiel, Germany) that was performed according to standard manufacturers protocol at NZFHRC, Nepal. Test validity was quantitatively ascertained by measuring the mean value (MV) of the optical density (OD) for the positive control, and the MV of the OD value for the negative control (NC) as per the manufacturers specifications. Test conditions were validated when the OD of the positive control was $\geq$ 0.850 OD units, and the OD for the negative control is $\leq$ 0.400 OD units. Test results were interpreted (Positive–Negative) by calculating the ratio (S/P) of sample OD to mean OD of the positive control according to the following equation: $\text{S/P} = \frac{\text{OD}_{\text{sample}} - \text{MV OD}_{\text{NC}}}{\text{MV OD}_{\text{PC}} - \text{MV OD}_{\text{NC}}}$

A sample with S/P < 0.25 was considered negative (specific antibodies to CDV could not be detected). According to protocol, samples with an S/P $\geq$ 0.25 was considered positive (specific antibodies to CDV were detected).

PCR analyses were performed at CMDN. Viral RNA was isolated from blood stored in VTM using TRIzol® Reagent (Thermo Fisher Scientific, Waltham, USA) and was further purified using Directzol™ RNA MiniPrep Kits (Zymo Research, Irvine, USA). Research Transcription was performed separately using Invitrogen Superscript III First Strand Synthesis kit (Cat# 18080–051), The PCR for phosphoprotein (P) gene (429 bp) detection was performed according to protocol [46] using Platinum Taq DNA Polymerase (Invitrogen) and identified in 1.5% agrose gel (Merck, Kenilworth, USA).

Research methods involving domestic dogs and human participants were reviewed and approved by the University of Tasmania's Animal Ethics Committee (Ethics reference: A0017120), and Human Ethics Committee (Ethics reference: H17190), respectively. Permission to conduct this study was granted by the Manang Nyeshang VDC Mayor's Office (Permit no.: (074/075) 261, 2075/01/19). The Manang Nyeshang VDC Mayor's Office, also referred to as "Manang Municipality Mayor's Office", is a government institution that oversees 14 villages, including the 10, that are within the study area. In addition, government representatives from each village committee were consulted and engaged in this study.

## Modelling

We used an information theoretic approach to multi-model inference using Akaike Information Criterion [47] to explain variation in number of dogs per household, and CDV seroprevalence. Prior to model fitting, all pair-wise associations among predictor variables (S1 Table) were evaluated using Spearman correlation analysis. No strong correlations ($\rho > 0.8$) were detected. Generalised linear models (GLMs) were constructed in the R statistical environment using RStudio version 1.1.383 (R core development team 2014). Confidence intervals were calculated using the Wilson Method.

We evaluate the effects of household factors (elevation, number of people per household, dog function, household income, and site) on the number of dogs per household, by using GLMs with Poisson error distributions. To investigate patterns of CDV seroprevalence in dogs, we used GLMs with binomial error distributions to assess the effect of site, age, household income, sterilised status, dog function, body condition score, sex, people per household, and roaming behaviour.

Of the ten villages from which we collected samples, two (i.e. Gyaru and Khangsar) were excluded from analyses owing to low sample size (there were only two dogs in each village). The remaining eight villages were clustered according to municipal association (i.e. managed by one governing body, with owners and their dogs frequently commuting between villages). The four village clusters used in our models were: Manang (Manang, Tanki Manang, and

Humde), Bhraka (Bhraka and Munji), Ngarwal, and Pisang (Pisang and Bhratang). This resulted in four village-clusters or single villages that were coded as sites in analysis.

In each case, the model averaged coefficients for the full model set were calculated. For the two response variables, the MuMIn package was used to run all possible model combinations of predictor variables for dogs per household, and all single and pairwise predictor variable combinations for CDV exposure (so models did not exceed data).

## Results

### Dog demography and husbandry

A total 71 dog-owning households (DOHH) responded to both the questionnaire and successfully volunteered their dogs for the serological study. Amongst DOHH, the number of dogs per household ranged between one and seven. Only five (7.0%) DOHHs had between three and seven dogs (that were not a part of a puppy litter), however the dogs in these households accounted for 23.9% of the total dog sample. The age and sex distribution, function, roaming behaviour, sterilised status, feed frequency and CDV seropositivity of dogs, both within village and across the study site, are summarised in Table 1.

Household income had the greatest relative importance in models predicting the number of dogs per household (Fig 1). However, as all coefficients overlapped zero, these variables were not useful predictors (refer to S1 File for complete model outputs).

Of the dogs surveyed, 74.7% were older than 12 months of age, and the male:female sex ratio was 2.9:1 or 74.7% male. Sterilisation rates were 45.3% for males and 50.0% for females (S1 Fig). The skew towards male dogs could be due to the deliberate selection by dog owners to limit population growth. Our findings that a low preference for sterilisation among dog

**Table 1. Distribution of dog variables across the study area and within each of the four village clusters.** Data from 71 individual dogs from distinct households were used in the analyses. Note that for all variables the sample size totals 71, except for CDV (n = 68).

| Variable | Categories | Bhraka | | | Manang | | | Ngawal | | | Pisang | | | Population | | |
|---|---|---|---|---|---|---|---|---|---|---|---|---|---|---|---|---|
| | | Male | Female | Total | Male | Female | Total | Male | Female | Total | Male | Female | Total | Male | Female | Total |
| Sex[a] | | 4 | 3 | **7** | 15 | | **15** | 9 | 4 | **13** | 25 | 11 | **36** | 53 | 18 | **71** |
| Age[b] (months) | 0–4 | | | **0** | 1 | | **1** | | | **0** | | 2 | **2** | 1 | 2 | **3** |
| | 5–12 | 2 | 1 | **3** | 4 | | **4** | 1 | | **1** | 5 | 2 | **7** | 12 | 3 | **15** |
| | >12 | 2 | 2 | **4** | 10 | | **10** | 8 | 4 | **12** | 18 | 9 | **27** | 38 | 15 | **53** |
| Function[b] | Guard | 2 | 2 | **4** | 1 | | **1** | 3 | 1 | **4** | 10 | 4 | **14** | 16 | 7 | **23** |
| | Herding | 1 | | **1** | 2 | | **2** | 1 | 1 | **2** | 1 | | **1** | 5 | 1 | **6** |
| | Pet | 1 | 1 | **2** | 12 | | **12** | 5 | 2 | **7** | 14 | 7 | **21** | 32 | 10 | **42** |
| Roaming[b] | Never | | | **0** | 5 | | **5** | 2 | 2 | **4** | 1 | | **1** | 8 | 2 | **10** |
| | Sometimes | 2 | | **2** | 4 | | **4** | 2 | 1 | **3** | 7 | 4 | **11** | 15 | 5 | **20** |
| | Always | 2 | 3 | **5** | 6 | | **6** | 5 | 1 | **6** | 17 | 7 | **24** | 30 | 11 | **41** |
| Sterilised[a] | Yes | 2 | 2 | **4** | 4 | | **4** | 7 | 3 | **10** | 11 | 4 | **15** | 24 | 9 | **33** |
| | No | 2 | 1 | **3** | 11 | | **11** | 2 | 1 | **3** | 14 | 7 | **21** | 29 | 9 | **38** |
| Feed frequency[b] | 2 per day | | 1 | **1** | 4 | | **4** | 3 | | **3** | 12 | 6 | **18** | 19 | 7 | **26** |
| | 3 per day | 4 | 2 | **6** | 11 | | **11** | 6 | 4 | **10** | 13 | 5 | **18** | 34 | 11 | **45** |
| CDV[c] | Positive | 2 | 3 | **5** | 14 | | **14** | 6 | 1 | **7** | 15 | 7 | **22** | 37 | 11 | **48** |
| | Negative | 2 | | **2** | 1 | | **1** | 2 | 2 | **4** | 9 | 4 | **13** | 14 | 6 | **20** |

[a] Determined through physical examination by a veterinarian.

[b] Determined through questionnaire.

[c] Determined with ELISA test.

| | Model | df | AICc | ΔAIC | Weight | D² |
|---|---|---|---|---|---|---|
| | *Number of dogs per household* | | | | | |
| 1 | ~ Household income | 2 | 142.21 | 0.00 | 0.21 | 0.15 |
| 2 | ~ Null | 1 | 142.22 | 0.22 | 0.18 | 0.00 |
| 3 | ~ Household income + People per household | 3 | 144.04 | 1.83 | 0.08 | 0.17 |

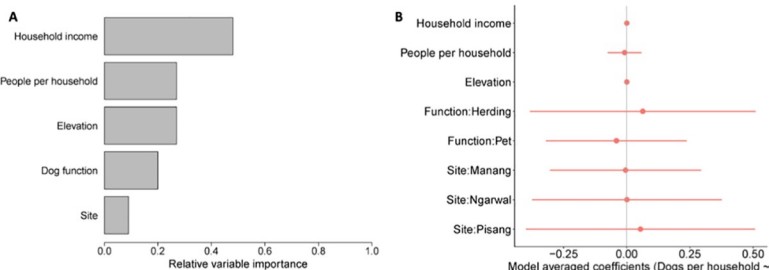

**Fig 1. Results for AIC analyses for the most parsimonious models (ΔAIC < 2) to predict the number of dogs per household, and (A) associated relative variable importance and (B) model averaged coefficients with 95% confidence intervals (red bars).** While models presented that household income had the greatest relative importance when predicting the number of dogs per household, variable coefficients overlapping zero suggest that these were not useful predictors.

owners was associated with a male-biased population seems to support this. Dogs fulfilled three primary functions: pet (59.2%), guarding (32.4%) and herding (8.5%) (S2 Fig). More than half (63.2%, n = 55) of dogs sampled were allowed to roam freely (Figs A-B in S3 Fig). Most (65.2%, n = 15) guard dogs and 52.4% (n = 22) of pet dogs were completely unrestrained.

More than 60% of dogs were fed at least three times a day (Figs C-D in S3 Fig). Most (60.6%, n = 43) respondents said they fed their dogs the same food eaten by the family. This primarily comprised rice, noodles, lentils, bread, barley, and occasionally meat. Approximately 35% of dogs had a body condition score (BCS) of 4 and 5 on a 9-point scale, which represent optimal condition (Figs E-F in S3 Fig). Clinical signs of CDV were not detected in any of the dogs sampled. At least 38 (53.5%) dogs had been vaccinated at least once in their lifetime, but only against rabies. Twenty seven dogs (38.0%) had been dewormed or visited a veterinarian in the past 12 months. Of the vaccinated dogs, 79.0% received free vaccinations from non-government organisations (i.e. Himalayan Mutt Project, Himalayan Animal Rescue Trust) and others had their dogs vaccinated at veterinary clinics in the city of Pokhara located approximately 200 km by off-road vehicle from the study region.

Questionnaire interviews revealed that 30.8% (n = 28) of dogs included in this study were sourced from outside of the study region. Owners reported moving their dogs frequently among villages, and observations reflected that it was customary for dogs to accompany their owners on foot, or within a vehicle on journeys to adjacent villages for work or social purposes. Owners also described quite frequent interactions between their dogs and native carnivores (S4 Fig and S2 Table). Fig 2 shows the source and destinations of human-assisted dog dispersal within the study region and connected urban centres. Pisang had the greatest proportion of dogs sourced from within the village, while Manang had the greatest proportion sourced from external villages. Dog owners from three of the four clusters reported sourcing them dogs from Kathmandu, the national capital. Villages within the study area were not only a destination, but also a source for dogs in villages outside of study area. Though it was not measured, packs of 4–6 dogs were frequently observed roaming within the villages.

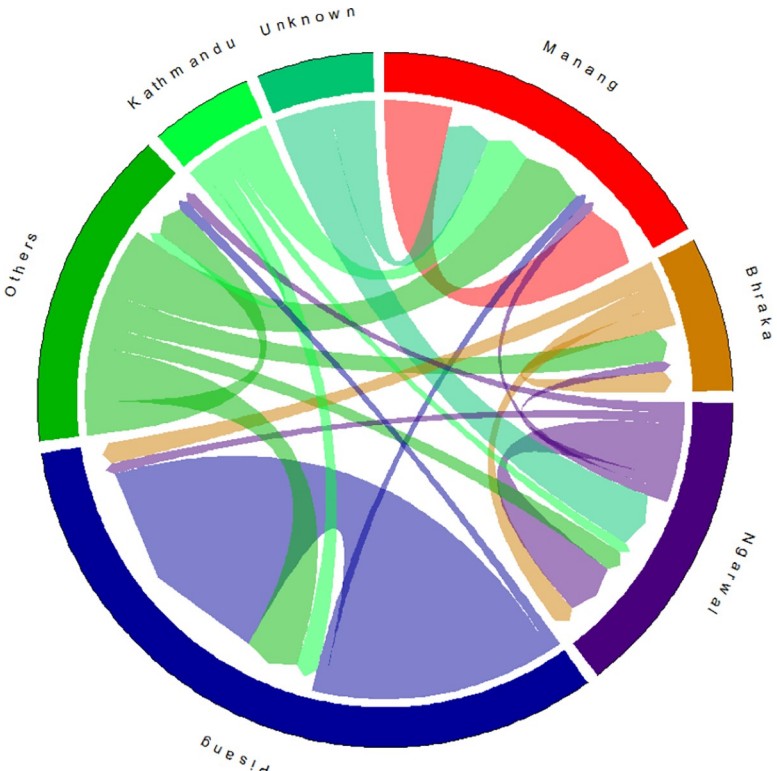

**Fig 2. The circular plot visualizes the dispersal flows of dogs with owners residing in upper study region.** The source and destinations of dogs are each assigned a colour and are represented by the circle's segments. The direction of the flow is encoded by the origin colour and the direction of the arrows. The volume of movement is indicated by the width of the flow. Because the flow width is nonlinearly adapted to the curvature, it corresponds to the flow size only at the beginning and end points. 'Others' represents locations excluded from the analyses: Chame, Tankchok, Timang, Tache, Sapche, Nar, Ghorka, and Humdre. Sectors are arranged in proximity to each other, i.e. Manang is the neighbouring village west of Bhraka, and Ngawal is the neighbouring village east of Pisang.

## CDV seroprevalence and P-gene detection in asymptomatic and non-vaccinated dogs

Sera from 68 dogs were analysed with ELISA assay to detect CDV antibodies, and 70.6% were seropositive (S3 Table). Fig 3A shows the percentage of samples that were positive for CDV antibodies at each site. All except one dog (94.1%) between 5–12 months were seropositive, and almost half (47.1%) of those above 12 months were seropositive. Serum from two pups (3–5 months), one of which was from a litter of five, were tested for antibodies and P-gene. Antibodies were detected in both pups, but both were negative for P-gene. One pup was sourced from Kathmandu and we were not able to test its mother. The mother of the second pup tested negative for antibodies. This suggests that her pup could have been exposed to CDV. While it is possible that the pup from Kathmandu could have carried maternal antibodies, it is also possible that the antibodies were due to exposure to CDV either in Kathmandu, during its journey or when it arrived at the village. A total of 58 (81.7%) dogs had sufficient blood were for PCR analysis, and of these, the target CDV P-gene was detected in seven (12.7%) individuals. Fig 3B shows the percentage of dogs that were positive for P-gene at each site.

Analyses of factors predicting seroprevalence produced a best-fitting model ($\Delta$AIC $\leq$ 2) that included site as a predictor variable, followed by age and then the other predictor variables

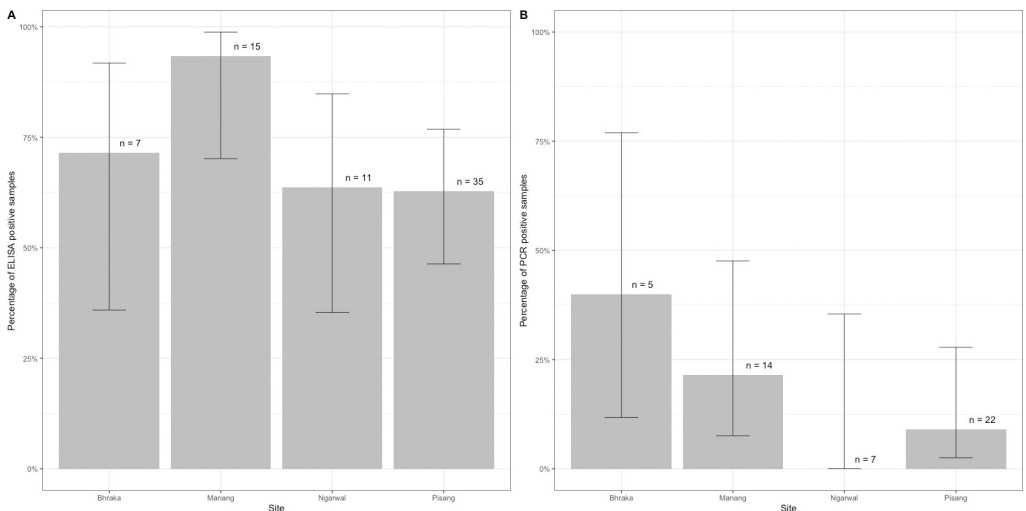

**Fig 3. (A) Percentage of ELISA-positive and (B) PCR-positive dogs at each site, and 95% confidence intervals.** Only samples that were ELISA-positive, and which had sufficient blood, were analysed for P-gene with PCR.

(Fig 4). However, as all coefficients overlapped zero, CDV seroprevalence was also not well predicted by any variable (Fig 4). Refer to S1 File for complete model outputs.

## Discussion

This study was the first survey of dog ownership and husbandry patterns in a Himalayan community, and the first survey of CDV exposure in Nepal Himalaya. The high CDV

| | Model | df | AICc | ΔAIC | Weight | D² |
|---|---|---|---|---|---|---|
| | *CDV sroprevalence* | | | | | |
| 1 | ~Age + Site | 6 | 62.34 | 0.00 | 0.36 | 0.31 |
| 2 | ~ Site | 4 | 63.79 | 1.45 | 0.18 | 0.22 |
| 3 | ~ Household income + Site | 5 | 63.97 | 1.63 | 0.16 | 0.25 |
| 4 | ~ Dog function + Site | 6 | 64.07 | 1.73 | 0.15 | 0.35 |
| 5 | ~ Site + Sterilisation status | 5 | 64.22 | 1.87 | 0.15 | 0.35 |

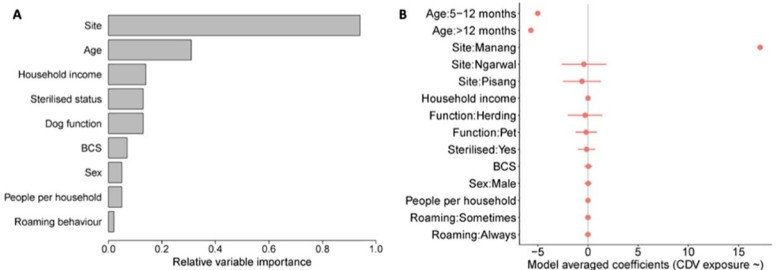

**Fig 4. Results for AIC analyses for the most parsimonious models (ΔAIC < 2) to predict the CDV seroprevalence, and (A) associated relative variable importance and (B) model averaged coefficients with 95% confidence intervals (red bars).** The confidence intervals for the variables "age" and "site" were relatively wide compared to the other variables and were removed. The 95% CI for Age: 5–12 months, Age: >12 months, and Site: Manang, were ±2.677$e^{+03}$, ±2.677$e^{+03}$, and ±1.944$e^{+03}$ respectively. Age group appeared to have some effect on CDV seroprevalence but coefficients overlapping zero indicate that these were not useful predictors.

seroprevalence among domestic dogs indicates that the virus is prevalent in the region. This could pose a risk to susceptible wildlife living in adjacent habitat. Free-roaming by dogs and high rates of human-assisted movement between rural and urban areas could facilitate this high prevalence of exposure to CDV.

Household- or village- level factors were not useful in predicting variation in the number of dogs per DOHH. This could suggest that the number of dogs per DOHH had little variation. The high proportion of dogs kept as pets was unexpected for this study region, which is a rural area where much of the population is engaged in pastoralism.

The finding that most dogs were free-roaming was expected and was common to studies conducted in similarly rural sites in developing countries [48–51]. These free-roaming dogs were also observed moving in packs both within and between villages, and owners reported that it was customary for their dogs to accompany them on journeys either on foot or by vehicle to adjacent villages for work or social visits. Owners reported sourcing their dogs from locations as far away as Kathmandu (Fig 2). Manang village had the largest proportion of dogs that were sourced from outside of the study area. A majority of the dogs in Pisang were sourced from within the village, with some dogs from Kathmandu. Despite the relatively small population of dogs in Bhraka, the village was a source of dogs in Pisang and Ngarwal. Similarly, Ngarwal was a source of dogs for villagers in Manang, Bhraka, and villages external to the study area. This human-assisted movement of dogs between urban and rural locations, and the local movement of dogs due to the unrestrained culture of dog keeping, results in a high degree of mixing. The high movement of dogs suggests that the dogs in the study area are effectively a single population. This would further explain why household- and village- level factors had no effect on abundance: local availability of food resources would not limit dogs able to roam freely to access resources from adjacent villages. That over a quarter of dogs in the study population originated from locations outside the study area (i.e. Chame, Tanchok, Timang, Sapche, Nar, Ghorka and Kathmandu) is important information for design of management interventions to limit the size of this dog population or manage health of dogs and disease risks to wildlife.

The age and sex of dogs was strongly skewed toward adults and males, respectively. The male-biased sex proportion was higher than the mean (62.6 ± 11.5) described by a meta-analysis of 85 rural dog populations from small rural villages or urban households in developing countries [6], and was consistent with other studies of rural dog populations [42,52–55]. The adult-skewed age ratio was also consistent with findings from the meta-analysis by Gompper [6].

The high proportion of dogs in ideal body condition recorded in this study was contrary to expectations for dogs in a developing country [56]. Sterilisation rates were also unusually high for a rural area and was well above the mean (11.6%) and median (6.2%) rates described by Gompper [6] for dogs generally, in which all rates above 10% were from studies in developed nations. The high rates of sterilisation might also underscore the value of collaborative efforts between local governments and non-profit organisations (i.e. Himalayan Mutt Project, Himalayan Animal Rescue Trust, etc.) to create access to sterilisation services.

Seroprevalence of CDV among unvaccinated and asymptomatic owned dogs was above the levels reported in comparable studies [57–61]. Few predictors of CDV were found, and the high degree of movement and mixing of domestic dogs could explain this. Manang village had the largest proportion of dogs that were CDV positive and sourced from outside the village. A similar study by Belsare and Gompper [10] reported that the rate of CDV incidence was positively influenced by the random introduction of an exposed dog. The high level of mixing could explain why we see little influence of household- or village-level variables on the exposure of individual dogs to CDV. The highly contagious nature of CDV, and extensive

movement of dogs within the study area could suggest that the transmission rate of CDV was high in study region. Future studies to investigate social-network and dog-wildlife interactions using spatial technology and remote camera trapping could help elucidate the mechanisms by which CDV becomes prevalent among domestic dogs. Such a study could also give insight to the role of human-assisted dog movement in facilitating pathogen transmission.

No dogs presented clinical signs of CDV. However, CDV is known to replicate rapidly in dogs with weakened immune systems [62], and potential stress caused by a stochastic event such as a natural disaster affecting communities in the region could cause CDV recrudescence [63] in the dog population, with ramifications on sympatric wildlife.

Residents reported observing direct interactions between dogs and wild carnivores, suggesting potential for pathogen transmission between dogs and susceptible wild carnivores. Given the high prevalence of CDV in dogs, it is conceivable that spillover might already have occurred. In the Annapurna Conservation Area, where wildlife conservation is an objective, the management of domestic dogs should be considered when developing and implementing wildlife conservation plans. There is also the risk of hybridisation between dogs and wild canids [64,65]. To address this, measures to promote vaccination and limit number of dogs per household and their roaming behaviour could be implemented. It would also be valuable for further research to investigate CDV exposure in sympatric wild carnivores.

## Conclusion

There is high CDV prevalence among dogs in Nepal's Annapurna Conservation Area, and spillover of disease from dogs to native wild carnivores is a potential threat to susceptible species. An outbreak among wild carnivores in the area, which includes endangered species, could cause sudden and rapid mortalities that could impact tourism and conservation objectives. A mass vaccination of domestic dogs could avoid the impact of an outbreak to the local economy, safeguard the wellbeing of dog owners, and buttress conservation objectives. The small population of dogs in the study area, most of which have owners, presents an opportunity to prevent the spread of CDV in wildlife that could come into contact with domestic dogs. The high susceptibility of dogs below the age of one year suggests that population control to reduce the number of susceptible individuals could be a cost-effective and a long-term strategy in preventing CDV circulation.

## Supporting information

**S1 Table. Variables relevant to CDV exposure that were selected to be investigated through the household questionnaire, and to be included in the model selection framework.** Variables were clustered into two levels–household, and dog. Based on the published literature, aspects of the household, and dog level were included for assessment in the questionnaire. (PDF)

**S2 Table. Variety and number of livestock that were reported killed by wild predators and domestic dogs within a 12-month period.** Numbers reflect reported livestock deaths associated with the corresponding predator. One rancher reported losing a total of 35 goats to a four different predators, but was unable to separate the number of livestock deaths associated with each predator. His report has been excluded from the table below. Also excluded from the table is the report by one rancher who lost more than 100 goats. The "Unknown" category describes reported livestock deaths associated with a predator attack of indiscernible identity. (PDF)

**S3 Table. Raw ELISA test results.** The table shows the S/P ratio that was derived from calculations as described in the methods, and the corresponding result.
(PDF)

**S1 Appendix. Questionnaire.** The complete questionnaire that was administered to residents in the study area.
(PDF)

**S2 Appendix. Dog sero-survey form.** The data sheet that was used to describe dog-level variables. Skin, body condition and BCS were ascertained by the same veterinarian throughout the study. Age was also ascertained by the veterinarian if such information was not available from the dogs' owner.
(PDF)

**S1 Fig. Sex and sterilised status of owned dogs in (A) study area, and (B) within each village cluster.** B: Bhraka (n = 7), M: Manang (n = 15), N: Ngawal (n = 13), P: Pisang (n = 36). Numbers in bars represent number of observations. Sterilised status across the study area was relatively evenly distributed between males and females. However, note the completely male-biased population in Manang and the low rate of sterilisation.
(PNG)

**S2 Fig. Function of owned dogs as described by their owners in (A) study area, and (B) within each village.** Numbers in bars represent number of observations. Manang and Pisang had the greatest proportion of pet dogs, while Bhraka had the greatest proportion of dogs kept for utility.
(PNG)

**S3 Fig. Roaming restrictions (A-B), feed frequency (C-D) and body condition of dogs (E-F) in the study area (A,C,E) and within each village cluster (B,D,F).** More than half of all dogs with owners were allowed to roam freely (A). Pisang and Bhraka had the largest proportion of free-roaming dogs (B). Most owned dogs were fed three times a day (C), except in Pisang where half the dog population was fed twice a day. Most dogs had an optimal BCS (E). Bhraka and Pisang had the largest proportion of dogs with ideal BCS, while Ngarwal had the largest proportion with a BCS <4. No dogs were scored 9. Numbers within the bars represent the number of observations.
(PNG)

**S4 Fig. Wild and native carnivores observed within the property boundaries of households interviewed.** Golden jackal: *Canis aureus*, and red fox: *Vulpes vulpes*, were the two most commonly sighted predators.
(TIF)

**S1 File. Complete model outputs.** The complete model outputs from the MuMIn package that was used to run all possible model combinations of predictor variables for dogs per household, and all single and pairwise predictor variable combinations for CDV exposure.
(PDF)

## Acknowledgments

The field work component of this study was funded by an Early Career Grant from the National Geographic Society. G. Ng, J. Kee, S. Tharm, and Z. Chua provided veterinary support for which we are grateful. Nepal-based partners, Animal Nepal, and Third Pole

Conservation made field work possible. Special thanks to the villagers of Manang who graciously completed the questionnaires and for allowing us to take samples from their dogs.

## Author Contributions

**Conceptualization:** Debby Ng, Scott Carver, Christopher N. Johnson.

**Data curation:** Debby Ng.

**Formal analysis:** Debby Ng, Scott Carver, Christopher N. Johnson.

**Funding acquisition:** Debby Ng.

**Investigation:** Debby Ng, Saman Man Pradhan, Ajay Narsingh Rana.

**Methodology:** Debby Ng, Scott Carver, Christopher N. Johnson.

**Project administration:** Debby Ng, Mukhiya Gotame, Dikpal Karmacharya, Ajay Narsingh Rana.

**Resources:** Debby Ng.

**Supervision:** Debby Ng, Scott Carver, Christopher N. Johnson.

**Validation:** Debby Ng.

**Visualization:** Debby Ng.

**Writing – original draft:** Debby Ng.

**Writing – review & editing:** Debby Ng, Scott Carver, Dibesh Karmasharya, Saman Man Pradhan, Christopher N. Johnson.

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
