## [Decision Letter · Decision Letter 0]

18 Aug 2019

PONE-D-19-20232

Canine distemper in Nepal's Annapurna Conservation Area

PLOS ONE

Dear Ms Ng,

Thank you for submitting your manuscript to PLOS ONE. After careful consideration, we feel that it has merit but does not fully meet PLOS ONE’s publication criteria as it currently stands. Therefore, we invite you to submit a revised version of the manuscript that addresses the points raised during the review process.

In addition to comments made by Reviewer 1, would you please consider the following suggestions:

- Line 178: "Where there were litters of pups below 12 weeks, only one individual was sampled as exposure to CDV is likely identical for litter-mates". How many of these pups tested positive (ELISA and PCR) ? Has maternal antibodies transfer been documented for CDV ? Do we know anything about the duration of immunity and how this may have affected your results ? (assuming some pups tested positive) ?

- Line 184: "The tubes were refrigerated at 5 °C in the field before being delivered to the National Zoonoses and Food Hygiene Research Centre (NZFHRC) in Kathmamdu". Were all samples shipped at the same time, in the same batch ? Were samples frozen or kept refrigerated on there journey to Kathmamdu ? Given the distance between Manang and Kathmamdu, I'm assuming that this may take several hours. Additional information on sample conservation would be welcome.

- Line 186: "The latter was stored in a Viral Transport Medium". Please provide information on the type of medium.

- Line 228: "Permission to conduct this study was granted by the Manang Nyeshang VDC Mayor’s Office (Permit no.: (074/075) 261, 2075/01/19)". Have you obtained permission from other villages ? From a governmental institution in Nepal ?

- Line 330: "Sera from 68 dogs were analysed with ELISA assay to detect CDV antibodies, and 70.6% were seropositive". How many dogs were sampled, tested, and positive, in each location ? This is a bit unclear throughout the manuscript, but critical for the interpretation of the data as location seems to be the most important variable. This could be presented in a figure presenting the seroprevalence per location (with 95% confidence intervals).

- Line 332: "Blood samples from 58 (81.7%) dogs were sufficient for PCR analysis, and of these, the target CDV P-gene was detected in nine (12.7%) individuals". Same remarks as above. In addition, I'm guessing that you targeted only a small portion of the P-gene with PCR system (you may provide the size in the method section). Was sequencing done on the positive samples ? If any, it would be great to, at least, include a brief genetic analysis of the detected viruses: level of similarity between sequences, phylogenetic analysis, etc. Given your results support that there is no population structure for the host (dogs), I wonder how this could affect the genetic diversity and spatial structure of CDV.

- Line 400: "Owners reported sourcing their dogs from locations as far away as Kathmandu". Any idea why ? Regarding the implementation of vaccination programs, wouldn't it be sustainable to target the major "dog sources", as detected in your study, rather than performing mass vaccination in remote locations ?

- Line 424: "The high proportion of dogs in ideal body condition recorded in this study was contrary to expectations for dogs in a developing country". Can this have affected your PCR detection rate ? Were there healthy dog shedding CDV in your study ? Might be interesting to have a look at the potential effect of dog body condition on viral shedding ?

- Finally, I also agree with Reviewer 1 that the dog-wildlife-livestock interaction part may be removed from the manuscript (or presented in the supplementary material section ). More information on the number of participant reporting predator would be needed, and potential biases affecting the response may need to be considered (do they own livestock ? amount of time spent watching for predators ? etc.). Types of interactions between dogs and each type of predators would also need to be considered (direct, indirect, allowing transmission of CDV or not, etc.).

We would appreciate receiving your revised manuscript by Oct 02 2019 11:59PM. To enhance the reproducibility of your results, we recommend that if applicable you deposit your laboratory protocols in protocols.io, where a protocol can be assigned its own identifier (DOI) such that it can be cited independently in the future. For instructions see: http://journals.plos.org/plosone/s/submission-guidelines#loc-laboratory-protocols

We look forward to receiving your revised manuscript.

Kind regards,

Camille Lebarbenchon

Academic Editor

PLOS ONE

Journal Requirements:

1. Please include additional information regarding the survey or questionnaire used in the study and ensure that you have provided sufficient details that others could replicate the analyses. For instance, if you developed a questionnaire as part of this study and it is not under a copyright more restrictive than CC-BY, please include a copy, in both the original language and English, as Supporting Information.

2. We note that [Figure(s) 1] in your submission contain [map/satellite] images which may be copyrighted. All PLOS content is published under the Creative Commons Attribution License (CC BY 4.0), which means that the manuscript, images, and Supporting Information files will be freely available online, and any third party is permitted to access, download, copy, distribute, and use these materials in any way, even commercially, with proper attribution. For these reasons, we cannot publish previously copyrighted maps or satellite images created using proprietary data, such as Google software (Google Maps, Street View, and Earth). For more information, see our copyright guidelines: http://journals.plos.org/plosone/s/licenses-and-copyright.

1.    You may seek permission from the original copyright holder of Figure(s) [1] to publish the content specifically under the CC BY 4.0 license. 

4. Please amend the manuscript submission data (via Edit Submission) to include authors Scott Carver, Mukhiya Gotame, Dibesh Karmasharya, Dikpal Karmacharya, Saman Man Pradhan, Ajay N. Rana, Christopher N. Johnson

Reviewers' comments:

Reviewer's Responses to Questions

**Comments to the Author**

1. Is the manuscript technically sound, and do the data support the conclusions?

Reviewer #1: Partly

Reviewer #2: Yes

2. Has the statistical analysis been performed appropriately and rigorously? 

Reviewer #1: No

Reviewer #2: Yes

3. Have the authors made all data underlying the findings in their manuscript fully available?

Reviewer #1: Yes

Reviewer #2: Yes

4. Is the manuscript presented in an intelligible fashion and written in standard English?

Reviewer #1: Yes

Reviewer #2: Yes

5. Review Comments to the Author

Reviewer #1: I have two major concerns with the research methods described in this paper.

1) Most of the dogs surveyed in this study were free-roaming, and the authors state that this was not an unexpected finding (Line 394). Indeed, a vast majority of dogs in Nepal (and the rest of the Indian subcontinent) are free-roaming, regardless of the ownership status. Furthermore, in these regions, the concept of dog ownership differs from that of the western world - dogs are loosely owned (village dogs/community dogs) and are not subject to any healthcare interventions like vaccination/deworming/neutering. In this context, the authors should explain how the 71 dogs they have included in this study ('belonging' to dog owning households) are different from the village dogs not included in this study. How representative these 71 dogs are of the larger village dog population? Can data obtained through convenience sampling (as per the information provided by the authors Lines 122-134) be extrapolated to infer about the regional dog population (e.g. Line 384 "The high CDV seroprevalence among domestic dogs...")?

The modeling/analysis used here also needs further explanation - for instance, Line 230: "..using Akaike Information Criterion (47) to explain variation in number of dogs per household and CDV seroprevalence": if 86% of the 'owned' dogs are free-roaming (Table 1), is it necessary to explore for factors like number of dogs per household to explain CDV seroprevalence? You are setting up a weak argument, and using statistical analysis/modeling to refute it. See L432 in your manuscript: "Few predictors of CDV were found, and the high degree of movement and mixing of domestic dogs could explain this."

2) The dog-wildlife-livestock interaction part in this manuscript is methodologically problematic (based on information derived from questionnaire surveys where the respondent perceptions and recall biases affect the response) and should be removed form this manuscript.

Minor comments: (L386) "Seropositive dogs reportedly interacting with wild carnivores suggests that this dog population could pose a risk of transmission of CDV to wild carnivores." Actually seropositive dogs do not play any role in the transmission of CDV - they are immune due to prior exposure. Dogs recovering from natural infection due to CDV develop a lifelong immunity to this pathogen (see Schultz, R., Thiel, B., Mukhtar, E., Sharp, P. & Larson, L. Age and long‐term protective immunity in dogs and cats. J. Comp. Pathol. 2010.142, S102–S108) and such dogs do not play any current or future role in the transmission of CDV. Reconsider line 469 as well: if dogs are exposed to CDV in young age, the survivors are actually immune (for life) and play no role in CDV transmission.

Provide confidence intervals wherever you report seroprevalence (e.g. L328)

L446: "No dogs presented clinical signs of CDV. The healthy body condition of most dogs could

have enabled them to resist the effects of disease (63)." Remove this as it is conjectural and not supported by your data.

Line 458: "To address this, measures to promote vaccination and limit number of dogs per household and their roaming behaviour could be implemented. " Do you think such recommendations are implementable?

Reviewer #2: I found this manuscript to be well written, clear and concise. This is novel and original research. The statistics are detailed and appropriate for the study. The design of the study is more than adequate and answers the questions posed by the investigators as to the epidemiology of an important viral pathogen in domestic dogs.

The conclusions are valid and supported by the data. The results and discussion section explain adequately how these conclusions were reached. The article is organized well and illustrates clearly the importance of this pathogen and the potential for its threat to wildlife in the region.

The study was done in an ethical manner and appears to have complied with all requirements for its performance. In addition, the raw data is available for scrutiny.

The Introduction reviews the published literature and presents it in a suitable manner for posing the questions asked. The methodology is adequately explained and appears to be complete and inclusive of the necessary information. In the Results section, Table 1 logically presents the data for assessment by the reader.

The only concern I have is the depth of the statistical analyses make assessment of the study and its conclusions a challenge for those unfamiliar with the methods described. Having said that, I believe the authors have adequately explained this methodology as outlined in the requirements.

---

## [Author Response · Author response to Decision Letter 0]

15 Oct 2019

Editors comments:

- Line 178: "Where there were litters of pups below 12 weeks, only one individual was sampled as exposure to CDV is likely identical for litter-mates". How many of these pups tested positive (ELISA and PCR) ? 

R: We have included additional information at L347: “All except one dog (94.1%) between 5-12 months were seropositive, and almost half (47.1%) of those above 12 months were seropositive. Serum from two pups (3-5 months), one of which was from a litter of five, were tested for antibodies and P-gene. Antibodies were detected in both pups, but both were negative for P-gene. One pup was sourced from Kathmandu and we were not able to test its mother. The mother of the second pup tested negative for antibodies. This suggests that her pup could have been exposed to CDV.”

- Has maternal antibodies transfer been documented for CDV ? Do we know anything about the duration of immunity and how this may have affected your results ? (assuming some pups tested positive) ?

R: Maternal antibody transfer for CDV has been reported in dogs and can be present for up to 8 weeks. While it is possible that the pup from Kathmandu could be carrying maternal antibodies, because our veterinarian estimated its age at between 10-12 weeks, it is also possible that the antibodies were due to exposure to CDV either in Kathmandu, during its journey or when it arrived at the village. The decision to collect one sample from each litter of pups was based on the purpose of statistical analyses; because litter-mates are not independent replicates, and due to the nature of CDV being a highly-infectious pathogen that can be transmitted by direct contact, including all individuals of a litter in the analyses would incur pseudo replication. As there was no evidence to conclude that the antibodies detected in the Kathmandu-sourced pup were maternal, and because the pup born in the village had a mother that had no antibodies, we do not have reasons to conclude that the presence of maternal antibodies have affected our results and analyses. It is also established in the literature, that unvaccinated pups are a susceptible age group because they are immunologically naive. Therefore, our recommendations include that birth control can be used as a strategy to manage the circulation of CDV. We have included additional information in L353: “While it is possible that the pup from Kathmandu could have carried maternal antibodies, it is also possible that the antibodies were due to exposure to CDV either in Kathmandu, during its journey or when it arrived at the village.”

- Line 184: "The tubes were refrigerated at 5 °C in the field before being delivered to the National Zoonoses and Food Hygiene Research Centre (NZFHRC) in Kathmamdu". Were all samples shipped at the same time, in the same batch ? Were samples frozen or kept refrigerated on there journey to Kathmamdu ? Given the distance between Manang and Kathmamdu, I'm assuming that this may take several hours. Additional information on sample conservation would be welcome.

R: All samples were shipped at the same time, in the same batch. During transport, samples were stored in an insulated box with refrigerant gel-filled freezer blocks. Samples were frozen upon reaching NZFHRC. Cornell University’s College of Veterinary Medicine advises that serum samples for antibody tests are stable at room temperature for several days. Preliminary assessments also showed that 16 of the 17 samples retrieved from our first day in the field tested positive for antibodies, and four samples from the first day were positive for P-gene. Therefore, we are confident that conservation methods did not compromise sample integrity. We have added additional information in L188: “All tubes were refrigerated at 5 °C in the field before being delivered to the National Zoonoses and Food Hygiene Research Centre (NZFHRC) in Kathmamdu, where the blood samples were frozen after being portioned for ELISA assay and PCR analysis.”

- Line 189: "The latter was stored in a Viral Transport Medium". Please provide information on the type of medium.

R: The VTM is commercially available. We have added the manufacturer information at Line 190: “The latter was stored in a RemelTM MicroTestTM M6TM Viral Transport Medium (VTM) before being delivered to the Centre for Molecular Dynamics (CMDN) in Kathmandu, Nepal.”

- Line 228: "Permission to conduct this study was granted by the Manang Nyeshang VDC Mayor’s Office (Permit no.: (074/075) 261, 2075/01/19)". Have you obtained permission from other villages ? From a governmental institution in Nepal ?

R: We had added the following information at L239: “The Manang Nyeshang VDC Mayor’s Office, also referred to as “Manang Municipality Mayor’s Office”, is a government institution that oversees 14 villages within Manang District. As all of the villages included in this study were within Manang District, only one government permit was necessary. In addition, the Chairperson of each Village Development Committee was consulted and engaged in this study.” Our efforts would not have been possible without their participation.

- Line 330: "Sera from 68 dogs were analysed with ELISA assay to detect CDV antibodies, and 70.6% were seropositive". How many dogs were sampled, tested, and positive, in each location ? This is a bit unclear throughout the manuscript, but critical for the interpretation of the data as location seems to be the most important variable. This could be presented in a figure presenting the seroprevalence per location (with 95% confidence intervals). Line 332: "Blood samples from 58 (81.7%) dogs were sufficient for PCR analysis, and of these, the target CDV P-gene was detected in nine (12.7%) individuals". Same remarks as above.

R: We have rephrased sections throughout the manuscript to add clarity about dogs that were sampled, tested, and positive in the ELISA and PCR analyses. We appreciate this comment as this made us realise that there was an error in the reporting of the sample sizes, but not in the analyses.

We have made amendments at L192-195: “Of the blood collected from 71 dogs, 68 (95.8%) had sufficient sera for ELISA assay. Of the ELISA-positive samples, 48 (67.6%) had sufficient blood for P-gene detection with PCR analysis.”

We have also modified the methods section providing details of the confidence intervals calculations in L250: “Confidence intervals were calculated using the Wilson Method.” 

We present Figure 4 to describe seroprevalence and PCR detections per location with 95% confidence intervals, and sample sizes. We have added L346: “Fig. 4A shows the percentage of samples positive for CDV antibodies at each site.”, and L357: “Fig. 4B shows the percentage of dogs that were positive for P-gene at each site.” And included the following figure and caption at L360:

Fig. 3 (A) Percentage of ELISA-positive and (B) PCR-positive dogs at each site, and 95% confidence intervals. Only samples that were ELISA-positive, and which had sufficient blood were analysed for P-gene with PCR.

- I'm guessing that you targeted only a small portion of the P-gene with PCR system (you may provide the size in the method section). Was sequencing done on the positive samples ? If any, it would be great to, at least, include a brief genetic analysis of the detected viruses: level of similarity between sequences, phylogenetic analysis, etc. Given your results support that there is no population structure for the host (dogs), I wonder how this could affect the genetic diversity and spatial structure of CDV.

R: Our PCR targeted a 429 nucleotide DNA fragment of the P-gene We have added the size of the gene fragment in Line 231: “The PCR for phosphoprotein (P) gene (429 bp) detection was performed according to protocol…”. 

Sequencing was not conducted on positive samples at this time. We appreciate your comment about investigating the relationship between host population structure and CDV genetic structure. This is in fact one of the research questions that we are currently targeting in our study area, however results are not available at the moment. 

- Line 400: "Owners reported sourcing their dogs from locations as far away as Kathmandu". Any idea why ? 

R: Unfortunately, we do not yet know why people source their dogs from Kathmandu. However, it could be related to seasonal migration by some residents from the villages to the capital city during the winter months, and the access created by new motor highways into mountain areas. We intend to investigate this aspect further in future studies. 

- Regarding the implementation of vaccination programs, wouldn't it be sustainable to target the major "dog sources", as detected in your study, rather than performing mass vaccination in remote locations ?

R: We agree that it is ideal and important to manage CDV in “source locations” such as Kathmandu. However, the population of dogs in Kathmandu is in the thousands and more than half of these have no owners making management actions logistically and economically difficult. That most of the dogs in these villages have owners, supports the implementation of a longterm management program. We have added L518: “The small population of dogs in the study area, most of which have owners, presents an opportunity to prevent the spread of CDV in wildlife that could come into contact with domestic dogs.” 

- Line 424: "The high proportion of dogs in ideal body condition recorded in this study was contrary to expectations for dogs in a developing country". Can this have affected your PCR detection rate ? Were there healthy dog shedding CDV in your study ? Might be interesting to have a look at the potential effect of dog body condition on viral shedding ?

R: While the proportion of dogs in ideal body condition was higher compared to other populations of dogs in developing countries, our sample population showed a normal distribution of body condition (S3: E-F, reproduced for your convenience below). As all dogs with owners, regardless of body condition, were included in our sample, we do not think body condition affected our PCR detection rate. All dogs in our study did not present clinical signs of CDV infection, however the detection of the P-gene in PCR assays support pathogen circulation in the blood stream. Owing to the small sample size of P-gene positive results (n=7), we were not able to derive any statistically meaningful conclusion about the effect of body condition on the PCR results.

- Finally, I also agree with Reviewer 1 that the dog-wildlife-livestock interaction part may be removed from the manuscript (or presented in the supplementary material section ). More information on the number of participant reporting predator would be needed, and potential biases affecting the response may need to be considered (do they own livestock ? amount of time spent watching for predators ? etc.). Types of interactions between dogs and each type of predators would also need to be considered (direct, indirect, allowing transmission of CDV or not, etc.).

R: We agree with the reviewers and, we have removed this section from the manuscript, and presented the figure and table in the Supplementary material instead. We have excluded mention of dog-wildlife-livestock interaction from the discussion, and revised the following lines in the manuscript:

L324: “Owners also described quite frequent interactions between their dogs and native carnivores (S4 Fig and S3 Table).”

L380: “This study was the first survey of dog ownership and husbandry patterns in a Himalayan community, and the first survey of CDV exposure in Nepal Himalaya.”

Reviewer #1 comments: 

- Explain how the 71 dogs they have included in this study ('belonging' to dog owning households) are different from the village dogs not included in this study. 

R: We have provided additional information in L125: “Dogs were considered “owned” if: the dog was fed daily by the same individual (or persons strongly associated with that individual, i.e. a family), and if the same individual provided for its basic healthcare (e.g. tending to wounds, fending off aggressive dogs, provided the dog a safe space to feed and rest, etc.).”

How representative these 71 dogs are of the larger village dog population? Can data obtained through convenience sampling (as per the information provided by the authors Lines 122-134) be extrapolated to infer about the regional dog population (e.g. Line 384 "The high CDV seroprevalence among domestic dogs...")?

R: We have provided additional information in L137: “The village dog population is well-represented by these 71 dogs as owned dogs comprised the majority of dogs across the 10 villages (only 15 dogs did not fit our criteria of having owners).

- The modeling/analysis used here also needs further explanation - for instance, Line 230: "..using Akaike Information Criterion (47) to explain variation in number of dogs per household and CDV seroprevalence": if 86% of the 'owned' dogs are free-roaming (Table 1), is it necessary to explore for factors like number of dogs per household to explain CDV seroprevalence? You are setting up a weak argument, and using statistical analysis/modeling to refute it. See L432 in your manuscript: "Few predictors of CDV were found, and the high degree of movement and mixing of domestic dogs could explain this."

R: Two models were developed. One to predict the number of dogs per household (Fig 2) and another to predict CDV seroprevalence (Fig 4). Number of dogs per household was not used to predict CDV seroprevalence (S3 File). We sought to investigate the number of dogs per household as previous studies have found this to be associated with disease risk (S1 Table). We have inserted a comma in line L246 to help make this distinction clear: “…number of dogs per household, and CDV seroprevalence.”

- The dog-wildlife-livestock interaction part in this manuscript is methodologically problematic (based on information derived from questionnaire surveys where the respondent perceptions and recall biases affect the response) and should be removed form this manuscript.

R: As mentioned above, we agree with the reviewers and have removed this section from the manuscript, and presented the figure and table in the Supplementary material instead. We have excluded mention of dog-wildlife-livestock interaction from the discussion, and revised the following lines in the manuscript:

L324: “Owners also described quite frequent interactions between their dogs and native carnivores (S4 Fig and S3 Table).”

L380: “This study was the first survey of dog ownership and husbandry patterns in a Himalayan community, and the first survey of CDV exposure in Nepal Himalaya.”

Minor comments: (L386) "Seropositive dogs reportedly interacting with wild carnivores suggests that this dog population could pose a risk of transmission of CDV to wild carnivores." Actually seropositive dogs do not play any role in the transmission of CDV - they are immune due to prior exposure. Dogs recovering from natural infection due to CDV develop a lifelong immunity to this pathogen (see Schultz, R., Thiel, B., Mukhtar, E., Sharp, P. & Larson, L. Age and long‐term protective immunity in dogs and cats. J. Comp. Pathol. 2010.142, S102–S108) and such dogs do not play any current or future role in the transmission of CDV. 

R: We have deleted the above sentence from the discussion, and added a new sentence in L382: “This could pose a risk to susceptible wildlife living in adjacent habitat.”

Reconsider line 469 as well: if dogs are exposed to CDV in young age, the survivors are actually immune (for life) and play no role in CDV transmission.

R: New born pups add to the proportion of susceptible individuals in a population. The pups could survive after going through an infectious phase, during which time it could play a role in transmission. If it recovers, it develops immunity. If it dies, viremia could pose a risk to other dogs or wildlife that come into contact with the carcass (CDV can persist in the environment for up to 14 days at 5degC (Watanabe, Y., H. Miyata, and H. Sato. "Inactivation of laboratory animal RNA-viruses by physicochemical treatment." Jikken dobutsu. Experimental animals 38.4 (1989): 305-311.); the average annual temperature in Manang is 6.5degC). Therefore, reducing the number of susceptible individuals through sterilisation remains a useful strategy to limit CDV circulation. 

Provide confidence intervals wherever you report seroprevalence (e.g. L328)

R: This has been addressed with the addition of Fig 4. Please see above.

L455: "No dogs presented clinical signs of CDV. The healthy body condition of most dogs could have enabled them to resist the effects of disease (63)." Remove this as it is conjectural and not supported by your data.

R: We have removed this statement, and made amendments to the paragraph at L493: “No dogs presented clinical signs of CDV. However, CDV is known to replicate rapidly in dogs with weakened immune systems (64), and potential stress caused by a stochastic event…”

Line 458: "To address this, measures to promote vaccination and limit number of dogs per household and their roaming behaviour could be implemented. " Do you think such recommendations are implementable?

R: Residents in Manang District have responded well to efforts by NGOs to provide neutering and vaccination services. Since the first neutering services were brought into these villages in 2014, 46% of owned dogs have been neutered. Since our study was completed in May 2018, the proportion of neutered dogs in the same villages has increased to 57%. Rabies vaccinations were delivered with each neutering effort. Five years ago, this number was zero. We have not citied this information in the manuscript as it is based on personal communication with local NGOs. Therefore, we have reasons to think that the promotion of CDV vaccinations could be similarly well received. We envisage that by presenting this information to the communities involved (community engagement events are planned for September 2019), dog owners and local government can be further encouraged to change the way they manage their dogs to improve One Health.

---

## [Editor Report · Decision Letter 1]

11 Nov 2019

Canine distemper in Nepal's Annapurna Conservation Area: Implications of dog husbandry and human behaviour for wildlife disease.

PONE-D-19-20232R1

Dear Dr. Ng,

We are pleased to inform you that your manuscript has been judged scientifically suitable for publication and will be formally accepted for publication once it complies with all outstanding technical requirements.

With kind regards,

Camille Lebarbenchon

Academic Editor

PLOS ONE
---

## [Editor Report · Acceptance letter]

18 Nov 2019

PONE-D-19-20232R1 

Canine distemper in Nepal's Annapurna Conservation Area – Implications of dog husbandry and human behaviour for wildlife disease 

Dear Dr. Ng:

I am pleased to inform you that your manuscript has been deemed suitable for publication in PLOS ONE. Congratulations! Your manuscript is now with our production department. 

With kind regards,

on behalf of

Dr. Camille Lebarbenchon 

Academic Editor

PLOS ONE